# Mosaic evolution in an asymmetrically feathered troodontid dinosaur with transitional features

Xing Xu[1,*], Philip Currie[2,*], Michael Pittman[3,*], Lida Xing[4], Qingjin Meng[5], Junchang Lü[6], Dongyu Hu[7] & Congyu Yu[8]

Asymmetrical feathers have been associated with flight capability but are also found in species that do not fly, and their appearance was a major event in feather evolution. Among non-avialan theropods, they are only known in microraptorine dromaeosaurids. Here we report a new troodontid, *Jianianhualong tengi* gen. et sp. nov., from the Lower Cretaceous Jehol Group of China, that has anatomical features that are transitional between long-armed basal troodontids and derived short-armed ones, shedding new light on troodontid character evolution. It indicates that troodontid feathering is similar to *Archaeopteryx* in having large arm and leg feathers as well as frond-like tail feathering, confirming that these feathering characteristics were widely present among basal paravians. Most significantly, the taxon has the earliest known asymmetrical troodontid feathers, suggesting that feather asymmetry was ancestral to Paraves. This taxon also displays a mosaic distribution of characters like *Sinusonasus*, another troodontid with transitional anatomical features.

[1] Key Laboratory of Vertebrate Evolution and Human Origins, Institute of Vertebrate Paleontology and Paleoanthropology, Chinese Academy of Sciences, Beijing 100044, China. [2] Department of Biological Sciences, University of Alberta, Edmonton, Alberta, Canada T6G 2E9. [3] Vertebrate Palaeontology Laboratory, Department of Earth Sciences, The University of Hong Kong, Pokfulam, Hong Kong. [4] School of the Earth Sciences and Resources, China University of Geosciences, Beijing 100083, China. [5] Beijing Museum of Natural History, Beijing 100050, China. [6] Institute of Geology, Chinese Academy of Geological Sciences, Beijing 100037, China. [7] Paleontological Institute & Key Laboratory for Evolution of Past Life in Northeast Asia, Ministry of Land and Resources, Shenyang Normal University, Shenyang 110034, China. [8] College of Life Sciences, Peking University, Beijing 100871, China. * These authors contributed equally to this work. Correspondence and requests for materials should be addressed to X.X. (email: xu.xing@ivpp.ac.cn).

The Middle-Upper Jurassic and Lower Cretaceous of western Liaoning and neighbouring areas have produced spectacular fossil remains of theropod dinosaurs, particularly of maniraptoran theropods, which are critical to our understanding of bird origins[1,2]. Among them are many fossils of troodontids, which are considered as the closest relatives of birds either on their own[3] or together with dromaeosaurids[4–8]. Reported troodontid species include *Sinovenator changii*[9], *Mei long*[10] and *Sinusonasus magnodens*[11] from the early Aptian Yixian Formation (the middle section of the Jehol Group)[12], *Jinfengopteryx elegans*[13] from the Hauterivian Dabeigou/Dadianzi Formation (the low section of the Jehol Group)[14], and *Anchiornis huxleyi*[7,15], *Xiaotingia zhengi*[6] and *Eosinopteryx brevipenna*[16] from the Oxfordian Tiaojishan Formation. Significantly, the discoveries of the latter four taxa provide direct evidence for the presence of feathers in troodontids, which shed new light on the early evolution of pennaceous feathers[6,7,13,16]. However, several recent phylogenetic studies have questioned the troodontid affinities of these species[3,6,17,18], and thus feathers have yet to be found in an unquestionable troodontid taxon.

Here we describe a new troodontid from the Yixian Formation of Baicai Gou, Yixian County, western Liaoning preserving large feathers along the forelimbs, hindlimbs and tail that have a distribution pattern similar to other basal paravians such as *Microraptor*, *Anchiornis* and *Archaeopteryx*[1,18–20]. Its discovery is highly significant in reconstructing both the skeletal and integumentary evolution of troodontids, and the more inclusive paravians.

## Results

### Systematic palaeontology.

Theropoda Marsh, 1881[21]
Coelurosauria Huene, 1920[22]
Maniraptora Gauthier, 1986[23]
Troodontidae Gilmore, 1924[24]
*Jianianhualong tengi* gen. et sp. nov.

**Etymology.** 'Jianianhua' (嘉年华), the Chinese company that supported this study; 'long' (龙), the Chinese Pinyin for dragon. The specific name honors Ms Fangfang Teng, who secured the specimen for study.

**Holotype.** DLXH 1218, a nearly complete skeleton with associated feathers (Fig. 1) housed at the Dalian Xinghai Museum.

**Locality and horizon.** Baicai Gou (白菜沟), Yixian County, western Liaoning, China; Lower Cretaceous Yixian Formation[12].

**Diagnosis.** A troodontid distinguishable from other taxa in possessing the following apomorphic features (*indicates autapomorphic feature): maxillary rostral ramus triangular in outline and relatively high dorsoventrally*; maxillary ascending process extending posterodorsally at a high angle (an angle of ∼45° to maxillary ventral margin)*; lacrimal with a long descending process sub-equal in length to anterior process; a prominent ridge along anterior edge of the lateral surface of the lacrimal descending process; a distinct fossa on the dorsal surface of the surangular close to its posterior end; axial neural spine with a convex dorsal margin, transversely thickened anterior margin, and posterodorsal portion expanding strongly posteriorly; long manual phalanx II-1 (slightly shorter than metacarpal III) with prominent proximoventral heel, large groove along the medial surface of more than proximal half of manual phalanx II-1*; highly elongated manual III-2 (slightly longer than metacarpal III)*; robust ungual phalanges (medial ungual proximal depth/ungual length

ratio >0.5); ilium with slightly concave dorsal margin in lateral view*; small medial lamina along ischial obturator process dorsal margin; metatarsal IV without prominent ventral flange*.

**Description.** DLXH 1218 is inferred to be an adult individual based on fusion features (for example, the centra and neural arches are completely fused to each other in all visible vertebrae)[25,26]. DLXH 1218 measures ∼100 cm in preserved skeletal body length, and is estimated to be ∼112 cm in total skeletal body length with a fully reconstructed tail. With a femoral length of ∼117 mm, its body mass is estimated to be 2.4 kg based on an empirical equation[27]. The estimated total skeletal body length and mass indicate that DLXH 1218 is similar in size to most other Jehol troodontids[9–11,13]. A succinct osteological description is given here (see Supplementary Table 1 for select measurements); for additional details, see Supplementary Note 1.

**Cranium.** The skull and mandible are in general well preserved (Fig. 2), but both premaxillae are missing and a few elements are preserved upside down, for example, the frontals. Otherwise the cranium indicates the presence of a sub-triangular cranial lateral profile, as in many other basal paravians. As in *Mei long*[10], it has a relatively short snout and highly expanded skull roof. The cranium is relatively small (the mandible is about 75% of femoral length), as in many other basal paravians.

The left maxilla is exposed, displaying a triangular outline. The rostral ramus, the anterior portion of the maxilla[28], is triangular in outline. It is slightly longer anteroposteriorly than dorsoventrally, unlike the rostral rami in other troodontids, which are considerably longer anteroposteriorly than dorsoventrally[29,30]. The ascending process extends posteriorly considerably beyond the anterior border of the antorbital fenestra. It extends posterodorsally at an angle of ∼45° to the ventral margin of the maxilla, an angle considerably greater than in other troodontids (for example, ∼36° in *Sinovenator*[9]; ∼6° in *Saurornithoides*[31]). Similar to *Byronosaurus*[32], a deep and narrow groove is present on the lateral surface of the jugal (ventral) ramus near the posterior end. There is a posteriorly deep jugal ramus as in derived troodontids[29].

The antorbital fossa is extensive, and sub-triangular in outline with a dorsally displaced anterior margin as in derived troodontids[29]. As in relatively derived troodontids[29], two large openings—the maxillary and antorbital fenestrae—are visible within the antorbital fossa in lateral view. In basal troodontids such as *Sinornithoides*, *Sinovenator* and *Sinusonasus* one additional opening—the promaxillary fenestra—is visible laterally[9,11,33] (as in the basal paravians *Anchiornis* and *Jinfengopteryx*[7,13]). The maxillary fenestra is enlarged as in other troodontids[29,34] (particularly derived ones[29,31,34]), is significantly elongated anteroposteriorly, and is displaced anteroventrally (its anterior and ventral borders are confluent rather than distant from those of the antorbital fossa, unlike basal troodontids such as *Sinovenator*).

Both lacrimals are exposed. The lacrimal is roughly a T-shaped bone, with an angle of ∼80° between the anterior and descending processes, and an angle of ∼105° between the posterior and descending processes. The long anterior process extends anteriorly beyond the anterior border of the antorbital fenestra, a feature that characterizes Troodontidae. At the junction of the descending and posterior process is a prominent lateral flange, as in all troodontids that preserve this anatomical region[35]. A similar structure is also present in most dromaeosaurids[35]. The long descending process is sub-equal in length to the anterior process, which is proportionally longer than in other

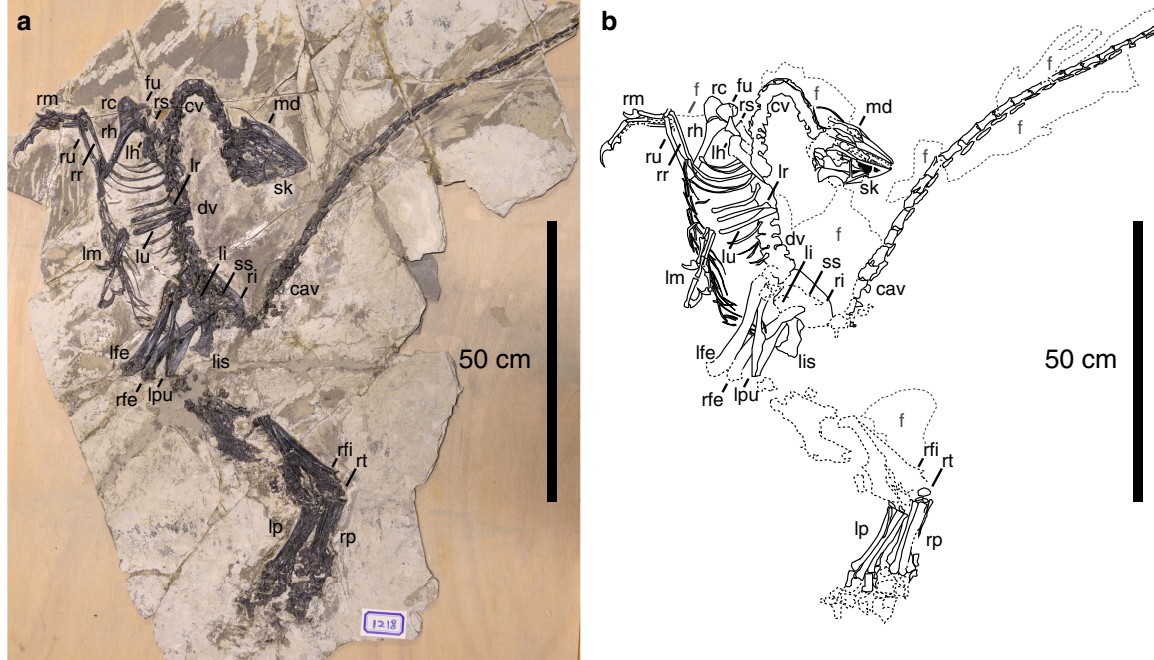

**Figure 1 | _Jianianhualong tengi_ holotype DLXH 1218.** (**a**) Photograph and (**b**) line drawing of the specimen. Scale bar, 50 cm. cav, caudal vertebrae; cv, cervical vertebrae; dv, dorsal vertebrae; fu, furcula; lfe, left femur; lh, left humerus; li, left ilium; lis, left ischium; lm, left manus; lp left pes; lpu, left pubis; lr, left radius; lu, left ulna; md, mandible; rc, right coracoid; rfe, right femur; rfi, right fibula; rh, right humerus; ri, right ilium; rm, right manus; rr, right radius; rs, right scapula; rt, right tibiotarsus; ru, right ulna; sk, skull; ss, synsacrum.

troodontids[29]. It curves anteriorly in lateral view as in many deinonychosaurs. A prominent ridge extends along the anterior edge of the lateral surface of the descending process, which is anteroposteriorly wider than transversely thick.

Both nasals are preserved but neither provides much morphological information. The nasal is short anteroposteriorly and relatively broad transversely as in _Mei long_[10]. Similar to other troodontids[29], there is a row of foramina along the lateral edge of this bone.

The prefrontals appear to be absent, but the possibility that small ones are present cannot be excluded.

Both frontals are nearly complete, and exposed in ventral view. The frontal is about 1.8 times as long anteroposteriorly as transversely wide. As in other troodontids[29], the postorbital process has a smooth transit from the orbital edge of the frontal.

The parietals are not well exposed, but were clearly fused to each other into one unit. They have transversely convex lateral surfaces, and have a weak sagittal crest.

The ventral margin of the anterior process of the jugal is thickened transversely due to a prominent ridge along the ventral border of the lateral surface of the jugal; the sub-orbital ramus is still deeper dorsoventrally than wide transversely. A similar ridge is also present in other troodontids[36]. As in _Sinovenator_ and other troodontids[30], the posterior process is short and deep, and has a lateral surface that is depressed into the articular facet for the quadratojugal.

The left quadrate is a posteriorly curved bone with a height of 22 mm. It is preserved upside down. The dorsal end of the quadrate is single-headed. The subtriangular pterygoid ramus is relatively small, narrow dorsally and wide ventrally, with the peak close to the ventral end of the quadrate. In most non-avialan theropods, the peak is more dorsally positioned.

A tri-radiate bone is identified as the left postorbital. The jugal process is the longest one of the three processes but does not seem as long as in other basal troodontids such as _Sinovenator_[30].

The mandible is subtriangular in lateral view, with a nearly straight dorsal margin and a convex ventral margin in lateral view. Both dentaries are preserved, and the left one is completely exposed. The dentary is a long and shallow bone, roughly triangular in lateral view as in other troodontids[29]. As in other troodontids[29], there is an anteriorly narrow and posteriorly wide groove on the lateral surface of the dentary.

The left surangular exposes its lateral side and the right one its medial side. The lateral surface of the surangular is dorsoventrally convex. The surangular is deep dorsoventrally, forming most of the lateral surface of the posterior portion of the mandible. There is a distinct fossa on the dorsal surface of the surangular close to its posterior end.

The lateral surface of the left angular is exposed, and the medial surface of the right one. The angular is a long and curved bone in lateral view. A robust, strongly dorsally curved anterior process of the angular excludes the dentary from the external mandibular fenestra as in _Sinovenator_ (for example, Institute of Vertebrate Paleontology and Paleoanthropology, IVPP V20378), some other troodontids, and to a lesser degree in some dromaeosaurids[30]; this probably represents a diagnostic feature for the Troodontidae or even the Deinonychosauria.

The medially exposed right splenial and the laterally exposed left splenial show that this bone is a sub-triangular plate, with its dorsal extremity near the dorsal margin of the mandible.

The medial side of the right prearticular suggests this is a typical ventrally curved maniraptoran one with a thin-bladed anterior portion and a somewhat rounded shaft-like posterior region. The dorsal margin of this bone is more strongly curved than the ventral margin.

The vertical columnar process of the right articular has a trapezoidal outline in medial view. Other morphological details are difficult to determine owing to their poor preservation, including those of the suspected left articular.

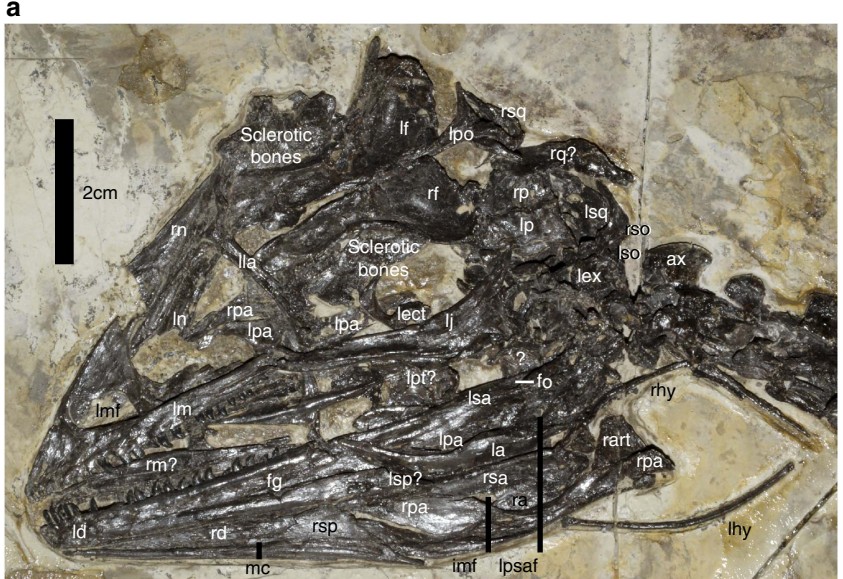

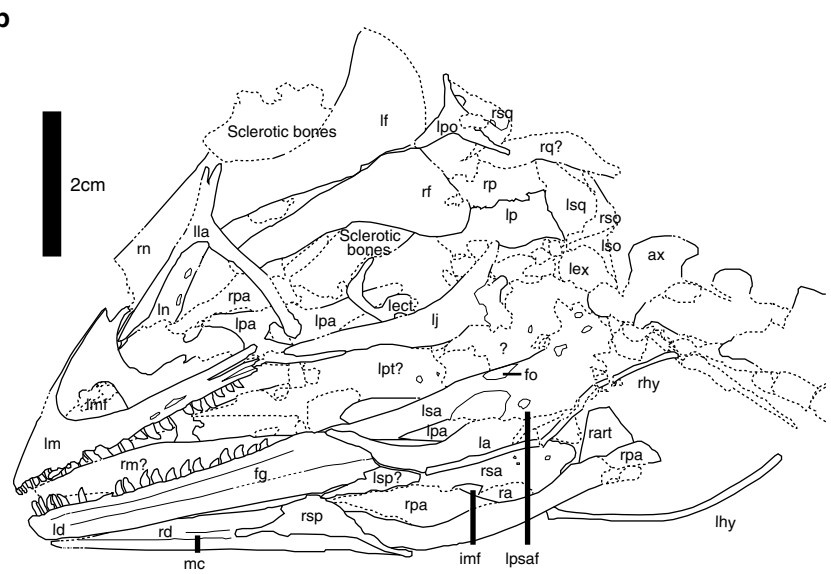

**Figure 2 | Cranium of *Jianianhualong tengi* holotype DLXH 1218.** (**a**) Photograph and (**b**) line drawing of the skull and mandible. Scale bar, 2 cm. ax, axis; fg, groove with foramina; fo, fossa; imf, internal mandibular fenestra; la, left angular; ld, left dentary; lect, left ectopterygoid; lex, left exoccipital; lf; left frontal; lhy, left hyoid bone; lj, left jugal; lla, left lacrimal; lm, left maxilla; lmf, left maxillary fenestra; ln, left nasal; lp, left parietal; lpt?, left pterygoid?; lpa, left prearticular; lpo, left postorbital; lpsaf, left posterior surangular foramen; lsa, left surangular; lsp?, left splenial?; lso, left supraoccipital; lsq, left squamosal; mc, Meckelian canal; ra, right angular; rart, right articular; rd, right dentary; rf, right frontal; rhy, right hyoid bone; rm?, right maxilla?; rn; right nasal; rp, right parietal; rpa, right prearticular; rq?, right quadrate?; rsa; right surangular; rso, right supraoccipital; rsp; right splenial; rsq, right squamosal.

The teeth preserved show an unevenly distributed (anterior teeth are closely packed, whereas the middle and posterior ones are more widely spaced), heterodont dentition (anterior teeth slender and leaf-like, whereas the middle and posterior teeth have stout, posteriorly curved crowns). Twelve teeth are preserved in the left maxilla and nine more are estimated to have been present; nineteen teeth are preserved in the left dentary and six more might have been present. Consequently, there are probably 21 maxillary teeth and 25 dentary teeth on each side of the jaw. The preserved maxillary teeth are strongly curved posteriorly and have relatively short and basally constricted crowns as in other troodontids. The dentary teeth are generally similar in morphology to the maxillary teeth: they have relatively short tooth crowns, lack serrations in anterior

teeth, and have relatively fine posterior serrations in middle and posterior teeth.

**Postcranium**. The vertebral column is nearly completely represented, but individual vertebrae are not well preserved (Fig. 3; Supplementary Note 1). Vertebral regions are roughly identifiable, but vertebral count is unknown. The cervicals are ~16 cm long, the dorsal region is ~17 cm long and the tail is ~54 cm long.

The axis has a unique neural spine with a convex dorsal margin in lateral view, an anterior portion that is transversely thickened and a posterodorsal portion that expands strongly posteriorly. The fifth to seventh cervicals are the longest ones of the cervical

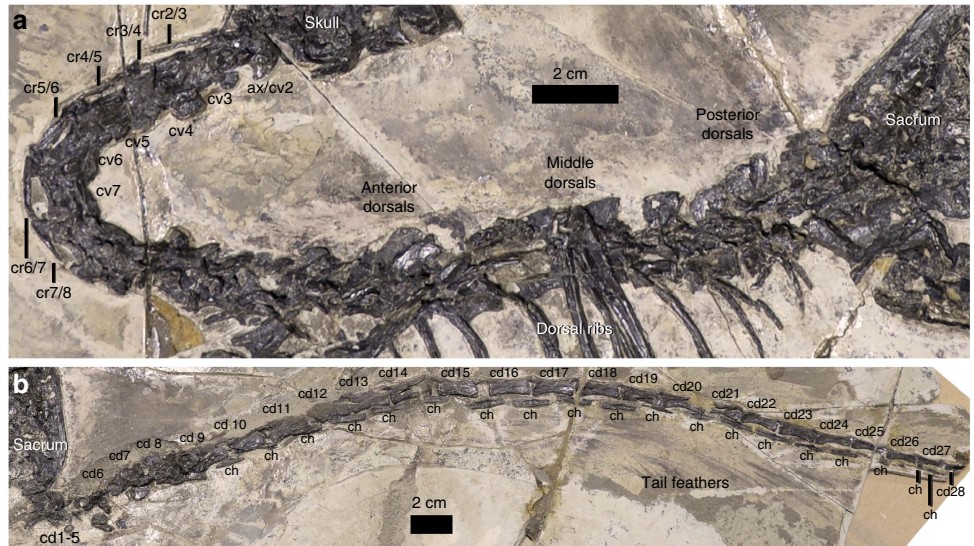

**Figure 3 | Vertebral column of *Jianianhualong tengi* holotype DLXH 1218. (a)** Presacral vertebrae. **(b)** Caudal vertebrae. Scale bar, 2 cm. ax, axis; cd, caudal vertebra; ch, chevron; cr, cervical rib; cv, cervical vertebra.

series. The latter two cervicals have complex pneumatic systems (foramina and fossae separated by ridges) on their anterolateral surfaces. Most cervical ribs, except the ones attached to the third and sixth cervicals, are at least partially fused to their corresponding vertebrae.

The dorsal series comprises anteroposteriorly short anterior dorsal vertebrae and long middle and posterior ones. The anterior dorsals lack distinct fossae or foramina on their central lateral surfaces, and probably so do the middle and posterior dorsals. Nine pairs of dorsal ribs are preserved. As in other troodontids[29], there are neither uncinate processes nor an ossified sternum. There are about 12 pairs of imbricated gastral segments preserved.

The sacral vertebrae are exposed due to the ventrally displaced left ilium. However, little morphological information is available due to the smashed nature of these bones.

An articulated series of 23 caudals is relatively well preserved, but the anterior caudals (~5 in number) are badly smashed and the distal caudals (~3 in number) are missing. Consequently, DLXH 1218 possibly has 31 caudals in total, making an estimated tail length ~3.9 times of femoral length. The tail possesses several typical troodontid traits including posterior chevrons with relatively blunt-ended anterior processes and slight bifurcated posterior processes (see Supplementary Note 1 for more details).

The shoulder girdle displays several features more similar to those of derived troodontids[29] (Fig. 4a). Both scapulae are preserved, with the left one incomplete and the right one partially exposed. The scapula is sub-equal in length to the humerus, and similar in thickness to the latter. Each scapula is preserved with its long axis angled at ~40° to the dorsal series as in most articulated specimens of non-avialan theropods. However, in articulated specimens of long-armed paravians, the scapula is nearly parallel to the dorsal series[30]. The acromion process continues smoothly from the scapular blade, extending anteriorly probably beyond the anterior limit of the glenoid fossa. Below the acromion process, the scapula is strongly depressed in lateral view. The scapular blade is strap-like, with an expanded distal end measuring ~130% of the minimum blade width.

The left coracoid is mostly missing and the nearly complete right is exposed in medial view. As in other derived maniraptorans[23], the coracoid is large and quadrangular, with the ratio of anteroposterior length to dorsoventral depth at the level of the

scapular suture ~1.3 (ref. 37), similar to those of the other pennaraptorans. The hooked postglenoid process of the coracoid is proportionally similar in size to that in *Sinornithoides*, but is longer than in *Sinovenator*[30].

The furcula is poorly preserved, but is robust, flattened in cross-section and U-shaped, as in *Mei*[38]. This contrasts with the more delicate furcula of *Sinornithoides*[38].

The left humerus is somewhat smashed, and mostly the anterior side of the right humerus is exposed. The humerus is ~70% of femoral length, and as in derived troodontids it is considerably more slender than the femur. The proximal portion of the humerus curves considerably posteriorly and medially. The inner tuberosity is long proximodistally and the deltopectoral crest is subtriangular in outline and short. It is ~20% of the total humeral length as in basal deinonychosaurs, but unlike the proportionally much longer ones seen in derived troodontids such as *Linhevenator*[36].

Both ulnae are exposed and relatively well preserved. The ulna measures 88% of humeral length. It is slightly bowed posteriorly, whereas in *Mei* it is more strongly bowed and in most other troodontids it is straight[10,23].

Both radii are relatively well preserved. The radius is slightly thinner than the ulna (76% of the shaft diameter). Unlike the straight radii in most other theropods, the proximal portion of the radius in *Jianianhualong* curves posteriorly.

The manus is typical of maniraptoran theropods, comprising three digits with a phalangeal formula of 0-2-3-4-0 (Fig. 4b). We follow the numbering for the wing digits of living theropods, as in most ornithological literature and some recent palaeontological studies[39,40]. The manus measures 112 mm in length, ~140% of humeral length and ~95% of femoral length.

The right wrist preserves three carpals, representing the semilunate carpal, distal carpal 4 and possibly radiale.

Metacarpal II is the most robust of the three metacarpals, and is ~45% of the length of metacarpal III. There is a distinctive ventral flange along the lateral edge of the proximal half of the ventral surface of metacarpal II, partially contributing to a large flat facet for contacting metacarpal III.

Metacarpal IV is a long element and it is much more slender than metacarpal III. It appresses closely to metacarpal III for its whole length.

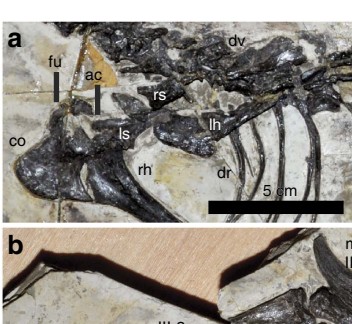

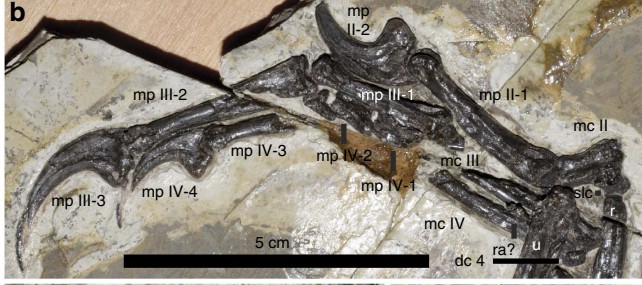

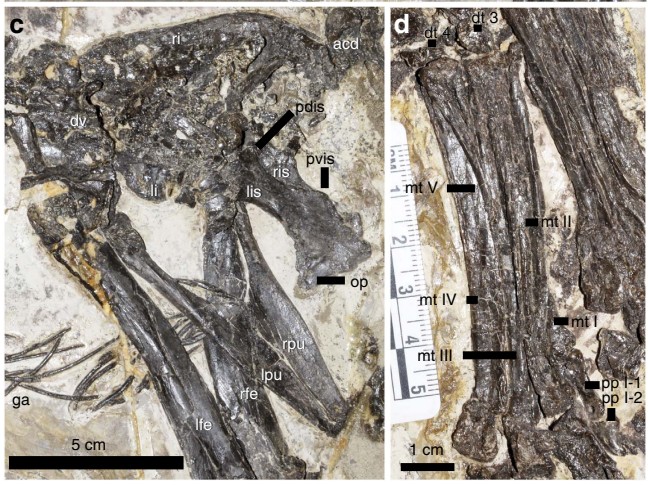

**Figure 4 | Non-vertebral postcranial skeleton of *Jianianhualong tengi* holotype DLXH 1218.** (**a**) Shoulder girdle, (**b**) right manus, (**c**) pelvis and (**d**) left pes. Scale bar, 5 cm except in **d**, where it is 1 cm instead. ac, acromial process; acd, anterior caudal vertebrae; co, coracoid; dc 4, distal carpal 4; dr, dorsal ribs; dt 3, distal tarsal 3; dt 4, distal tarsal 4; dv, dorsal vertebrae; fu, furcula; ga; gastralia; lfe, left femur; lh, left humerus; li, left ilium; lis, left ischium; lpu, left pubis; ls, left scapula; mc II to IV, metacarpals II to IV; mp II-1 to II-2, manual phalanges II-1 to II-2; mp III-1 to III-3, manual phalanges III-1 to III-3; mp IV-1 to IV-4, manual phalanges IV-1 to IV-4; mt I to V, metatarsals I to V; pp I-1 to I-2, pedal phalanges I-1 to I-2; pdis, dorsal posterior process of ischium; pvis, ventral posterior process of ischium; r, radius; ra?, radialae?; rfe, right femur; rh, right humerus; ri, right ilium; ris, right ischium; rpu, right pubis; rs, right scapula; slc, semi-lunate carpal; u, ulna.

Manual phalanx II-1 is a long and robust element. It is slightly shorter than metacarpal III. It bears a prominent proximoventral heel so that the proximal end is 2.3 times as deep dorsoventrally as the shaft immediately proximal to the distal end, which represents the thinnest portion of the bone. In lateral view, manual phalanx II-1 curves slightly ventrally. There is a prominent, wide groove along the medial surface of the proximal half of this phalanx.

Manual phalanges III-1 and III-2 are similar in general morphology to II-1, but lack the distinctive proximoventral heel and are straight in lateral view. The former further differs from II-1 in having shallower and more ventrally positioned collateral ligament pits. The highly elongated manual III-2 is slightly longer than metacarpal III, a feature only known in certain

ornithomimosaurs, and in several basal paravians such as *Xiaotingia* and *Yixianosaurus*[37].

Manual phalanx IV-1 is sub-equal in length to IV-2, but combined they are shorter than IV-3 as in most derived maniraptorans and ornithomimosaurs.

The three ungual phalanges are robust (ungual 2 is the most robust and has a proximal depth more than half its length), highly curved, laterally compressed, and with prominent flexor tubercles.

The pelvis is in general similar to those of basal troodontids such as *Sinovenator*[9]. It has a proportionally small ilium, a posteroventrally oriented pubis, and a short ischium with a distally positioned obturator process (Fig. 4c).

The right ilium is complete, but obscured by sacral vertebrae and the proximal caudals; the left ilium is missing most parts. The ilium is small in size in comparison to the femur (ilium/femur length ratio is ∼0.60), which is the condition in *Archaeopteryx* and basal dromaeosaurids[41]. In most other theropods, the ilium is more than 70% the length of the femur. The dorsal margin of the ilium is slightly concave in lateral view, unlike the straight or convex ones seen in many other theropods. The pubic peduncle is anteroposteriorly wide and deep, and is slightly posteriorly directed, with a large depression on its anterolateral surface representing the anteriorly and laterally positioned fossa for the M. cuppedicus muscle[42].

The left and right pubes are, as in other troodontids, appressed to form a broad, flat pubic apron. The dorsal edges along the midline meet and extend posteriorly to form a longitudinal ridge. The transversely hypertrophied pubic apron is characteristic of troodontids among theropods[43], and the midline ridge is at least also known in *Sinovenator*.

Both ischia are preserved. The ischium is a short plate, 48% of pubic length. Proximally, close to the iliac articulation, is a small dorsal process as in basal dromaeosaurids and some basal birds[41,44,45]. Distal to this process, the dorsal margin is slightly concave in lateral view, and then expands posteriorly to form a second dorsal process close to the distal end (Fig. 4c), as in *Archaeopteryx*[20]. A distally positioned obturator process is present as in basal deinonychosaurs and some basal birds, but it is relatively small, closer in size to that of *Archaeopteryx*[20]. There is a small lamina along the dorsal margin of the obturator process, a feature also known in some dromaeosaurids such as *Sinornithosaurus millenii* and *Buitreraptor*, although in these taxa this lamina is much larger.

Both femora are preserved, but neither preserves many morphological details (Fig. 4c). The femur is significantly bowed anteriorly as in most theropods[46]. The distal half of the femur is considerably more robust than the proximal half, a feature also known in *Anchiornis huxleyi*[15].

Both tibiotarsi are badly smashed and little morphological information is obtainable, which is also the case with the fibulae. Some badly smashed bones might be proximal tarsals.

Two disk-like distal tarsals are identified as distal tarsals 3 and 4 because they cover the proximal ends of metatarsals III and IV, respectively (Fig. 4d). The possibility that these two elements are partially fused to each other cannot be excluded.

Metatarsal I is a short, proximally tapered bone that attaches to the medial surface of metatarsal II about three-fifths of the way down the metatarsus (Fig. 4d). Metatarsal II is the most slender element of the three middle metatarsals and is also considerably shorter than the other two (metatarsal II/III length ratio 0.88). The distal end of metatarsal II is not ginglymoid as in other troodontids[46]. Metatarsal IV is the most robust element as in other troodontids[46], but a prominent ventral flange appears to be absent. The slender and rod-like metatarsal V is 40% of the length of metatarsal III.

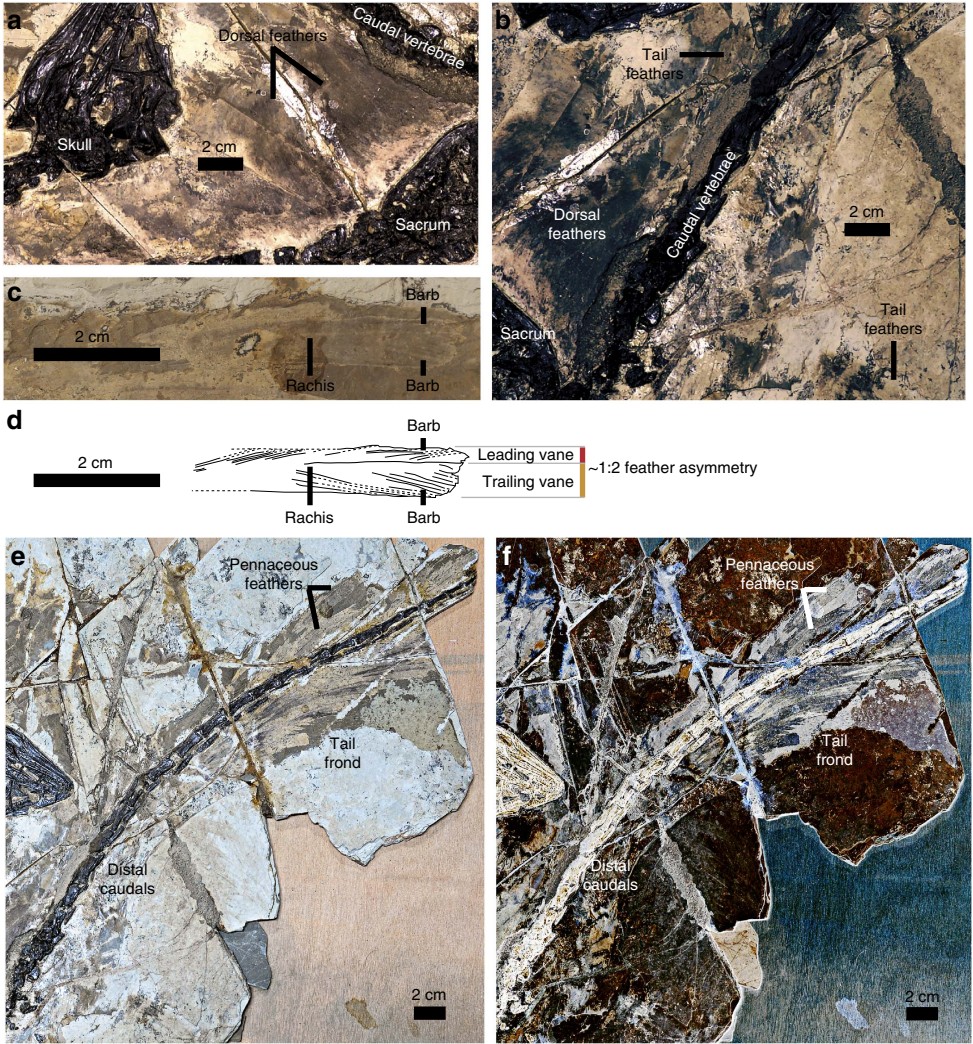

**Figure 5 | Plumage of *Jianianhualong tengi* holotype DLXH 1218.** (**a**) Feathers over dorsals, (**b**) feathers attached to anterior caudals, (**c**) asymmetrical tail feather, (**d**) line drawing of asymmetrical tail feather, (**e**) tail frond and (**f**) negative LSF image of tail frond. All scale bars, 2 cm.

Pedal digit I is proportionally long (phalanges I-1 and I-2 combined/metatarsal III length ratio 0.31; Fig. 4d), which is almost the same as in *Troodon* (CMN 8539). Distally it extends beyond the distal end of metatarsal III. Pedal phalanx I-1 is proportionally long (phalanx I-1/metatarsal I length ratio 1.9). The ungual is robust (proximal end depth/ungual length ratio ∼0.50) and moderately curved.

Pedal digit II is typical of a derived troodontid. Phalanx II-1 is ∼150% of the length of phalanx II-2, with a dorsoventrally deep proximal end. Pedal phalanx II-2 has a proximoventral heel, which has both considerable proximal and ventral extensions as in derived deinonychosaurs[47–50]. As in other troodontids and basal dromaeosaurids[51], the distal end extends moderately below and above the shaft indicating a wider rotation arc than in other theropods. The ungual of digit II is similar to those of some derived troodontids and dromaeosaurids[36,49] in being strongly curved, with a prominent flexor tubercle; it is also much larger than pedal phalanges II-1 and II-2 (for example, II-3/II-1 length ratio ∼1.4). In basal troodontids such as *Sinovenator*, the ungual of digit II is about the length of pedal phalanx II-1.

**Plumage**. Feathers are preserved along nearly the whole vertebral series, forelimb and hindlimb (Fig. 5). Although most of these feathers are large, few morphological details are preserved other than the ones associated with the caudal series. Laser-stimulated fluorescence (LSF) imaging did not reveal any hidden feather details (Fig. 4 of ref. 52), but made existing details easier to see.

Feathers ventral to the cervical series are at least 30 mm long. The ones dorsal to the dorsal and sacral vertebrae seem to be much longer than the cervical feathers, and the ones above the posterior dorsals and sacrals are ∼75 mm long. Large pennaceous feathers are preserved both dorsal and ventral to nearly the whole preserved caudal series as in *Jinfengopteryx*, *Archaeopteryx* and *Anchiornis*, forming a frond-like feathery tail, a primitive morphotype of tail feathering. These feathers are curved such that their distal edge is convex and their proximal edge is concave, unlike *Archaeopteryx* and possibly *Jinfengopteryx* too[18,20,53]. Most caudal feathers are difficult to measure, but the ones associated with the middle caudal vertebrae are ∼120 mm long and at least 10 mm wide. Interestingly, one lateral caudal feather has relatively strong asymmetry, with the trailing vane about twice as wide as the leading vane with barb angles of ∼10° and ∼15°, respectively (Fig. 5c,d). This feather and some other lateral caudal feathers appear to be distally blunt, with the distal portions even wider than the more proximal regions. The lateral caudal feathers of *Archaeopteryx* have an asymmetrical outline and rachis position, but its distal caudal feathers have a symmetrical,

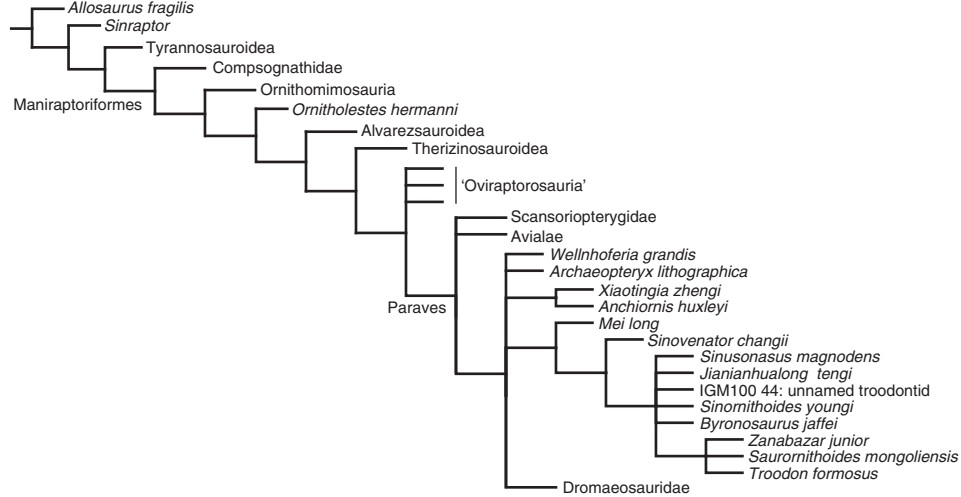

**Figure 6 | Strict consensus of 3,066 most parsimonious trees produced by phylogenetic analysis.** The strict consensus tree (tree length = 1,426, CI = 0.279) shows the systematic position of *Jianianhualong tengi* among the Coelurosauria.

rounded outline at the tip as well as a seemingly symmetrical rachis position[18].

Feathers are also preserved near the forelimb (humerus and ulna), but little morphological information is obtainable. Posterior to the preserved distal half of the tibia, there are also some poorly preserved feathers. Those near the middle portion of the tibia appear to be 70 mm in length, but few details are preserved.

The newly reported asymmetry in the tail feathers of *Jianianhualong* was incorporated into a parsimony-based ancestral state reconstruction of paravian arm and tail feather symmetry (see Supplementary Fig. 1; Supplementary Table 2 and Methods). This analysis revealed vane asymmetry as the ancestral condition for paravian arm feathers and asymmetrical tail feathers as the ancestral condition of a more inclusive paravian clade that excludes Scansoriopterygidae and Avialae. However, if the ancestral condition of scansoriopterygid and dromaeosaurid arm and tail feather symmetry is entered as being equivocal, then the reconstructed ancestral condition for paravian arm feather symmetry is also equivocal whereas it is asymmetrical for paravian tail feathers.

## Discussion

*Jianianhualong* shares with other troodontids many apomorphic features[29], which are probably troodontid synapomorphies. These features include a long anterior process of the lacrimal that extends anteriorly beyond the anterior border of the antorbital fenestra, a prominent lateral flange over the descending process of the lacrimal, a row of foramina along the lateral edge of the nasal, a postorbital process of the frontal that transitions smoothly from the orbital rim, a ridge close to the ventral edge of the jugal lateral surface, a triangular dentary in lateral view, an anteriorly narrow and posteriorly wide groove on the lateral surface of the dentary, a strongly curved, robust anterior process of the angular that contacts the surangular to exclude the dentary from the external mandibular fenestra, a relatively large number of teeth (25 + dentary teeth), uneven dentition distribution, heterodont dentition, dorsoventrally flattened chevrons with blunt-ended anterior processes and shallowly bifurcated posterior ones, and a pubic apron that is transversely broad and flat. Our numerical phylogenetic analysis places *Jianianhualong* in an intermediate position together with several species between the basalmost and derived troodontids (Fig. 6, see Methods).

The discovery of *Jianianhualong* provides direct evidence for the presence of pennaceous feathers in an unquestionable troodontid theropod. Previous studies indicate that *Anchiornis huxleyi*, *Eosinopteryx brevipenna*, *Jinfengopteryx elegans* and *Xiaotingia zhengi* have pennaceous feathers[6,7,13,16]. However, some recent studies have questioned the troodontid affinities of these species[3,6,17,18]. *Jianianhualong* is clearly a troodontid theropod as indicated by numerous troodontid features, and its discovery thus confirms the presence of feathers of modern aspect in this key theropod group. Furthermore, the presence of large feathers on the tail, forelimbs and hindlimbs in *Jianianhualong* confirms that these feathering characteristics were widely present in basal paravians[54]. This provides a robust troodontid basis for deepening understanding of the aerodynamic capabilities of this condition, beyond the gliding-related insights from the Jehol dromaeosaurid *Microraptor*[55,56].

The tail frond of *Jianianhualong* preserves an asymmetrical feather (as well as some suspected ones), the first example of feather asymmetry in troodontids. Feather asymmetry has been suggested to be closely correlated with flight capability[57,58], and although the functional implications of feather asymmetry have been debated[59,60], the appearance of asymmetrical vanes in flight feathers has been considered to represent one of the major events in feather evolution[1,19,61]. Previously, arm feathers with asymmetrical vanes have been reported only in the Microraptorinae, a dromaeosaurid subgroup, among non-avialan theropods[62,63]. The discovery of tail feathers with asymmetrical vanes in a troodontid theropod adds significant new information on feather evolution. Assuming the most parsimonious explanation in which feather asymmetry only evolved once, this first troodontid record suggests that feather asymmetry was ancestral to Paraves. If feather asymmetry evolved across Paraves convergently, perhaps a less likely scenario, it would still have originated amongst basal paravians. However, both scenarios indicate a higher degree of aerodynamic prowess among basal paravians than has been previously appreciated, potentially to the benefit of locomotion be it on the ground (possibly), up inclines or through the air[64]. In *Anchiornis huxleyi*, *Eosinopteryx brevipenna*, *Jinfengopteryx elegans* and *Xiaotingia zhengi*, the arm feathers have apparently symmetrical vanes on either side of the rachis, but this cannot be determined in *Jianianhualong*. In contrast, only the tail feathers of *Jianianhualong* are sufficiently preserved to determine their

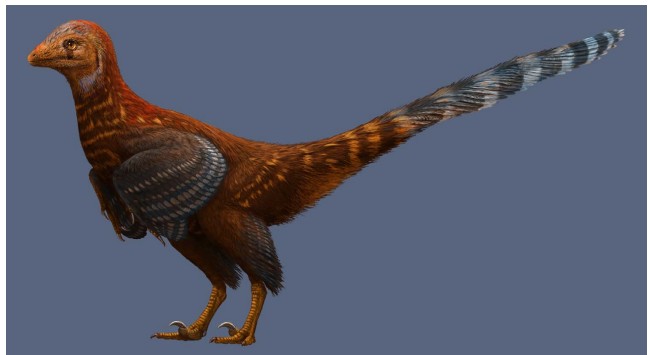

**Figure 7 | Life reconstruction of *Jianianhualong tengi*.** Life reconstruction of *Jianianhualong tengi* by Julius T. Csotonyi based on the holotype DLXH 1218.

vane symmetry[18,65]. Parsimony-based ancestral state reconstruction of paravian arm and tail feather symmetry supports a paravian ancestor with asymmetrical arm feathers (Fig. 6; Supplementary Fig. 1). The analysis also supports asymmetrical tail feathers as the ancestral condition of a more inclusive paravian clade that excludes Scansoriopterygidae and Avialae (Fig. 6; Supplementary Fig. 1). However, it should be noted that entering the ancestral scansoriopterygid and dromaeosaurid condition as equivocal for both the arm and tail feathers results in an equivocal reconstruction at the paravian node for arm feather symmetry, but reconstructs asymmetry for the tail feathers. Thus, future fossil finds and new information revealed from existing fossils using novel observational techniques such as LSF imaging (Fig. 5f; Figs 3 and 4 of ref. 52) will be crucial towards developing a robust understanding of the pattern and significance of paravian vane asymmetry evolution. Asymmetry in the width of the leading and trailing edge vanes (Fig. 5c,d) in the tail feathers of *Jianianhualong* appears to have enabled the animal to dynamically maintain feather stability, or pitch in airflow[66,67] (with the low angle of the leading edge barbs ($\sim 10°$) providing increased vane rigidity[68]). This aerodynamic capability may have benefited the locomotion of *Jianianhualong* on the ground and/or through the air, although in isolation these two aforementioned characteristics do not support either passive gliding or active, powered flight[60]. The tail feathers of *Jianianhualong* also have small trailing edge barb angles ($\sim 15°$), so are distinct from aerodynamically modern primary feathers[60]. It is worth noting that the subtriangular outline of the tail frond[69,70] and the wing-like slots between its outward-curving feathers may have been helpful towards drag reduction when the tail was used[69,70]. In having ancestrally asymmetrical vane-widths, the feathers of early paravians (Supplementary Fig. 1) may have been similarly selected upon for improvements to locomotion along the ground and/or through the air. However, the complex factors controlling asymmetrical development in living birds[71] suggests that this hypothesis requires further testing and development.

It is worth hypothesizing that *Jianianhualong* and other troodontids are likely to have tail feathers with asymmetrical vanes, but arm feathers with symmetrical vanes. This suggests that feather asymmetry first evolved in the troodontid tail and then appeared in other parts of the body. This hypothesis could potentially apply to paravians more generally, though not supported by our parsimony-based ancestral state reconstruction of paravian arm and tail feather symmetry (Supplementary Fig. 1). However, given the controversies over maniraptoran phylogeny, this significant feature deserves further investigation and is potentially important for our understanding of the

evolution of flight feathers from both morphological and functional perspectives.

The discovery of *Jianianhualong* adds to the diversity of the Jehol troodontids (Fig. 7). Previously, four troodontid species have been reported from the Lower Cretaceous Jehol Group: *Mei long*[10], *Sinovenator changii*[9], *Sinusonasus magnodens*[11] and *Jinfengopteryx elegans*[13]. *Jianianhualong* and *Sinusonasus* are more crownward than *Mei* and *Sinovenator*, and they display many features shared with more derived troodontids but absent in basal troodontids. This indicates the presence of a diversity of troodontids in the Early Cretaceous Jehol Group.

*Jianianhualong* possesses a mosaic of plesiomorphic and apomorphic morphological features throughout the skeleton, as in troodontids such as *Jinfengopteryx*[53]. However, in *Jianianhualong* this mosaic displays a distinct spatial organization, which suggests that character evolution occurred regionally rather than randomly throughout its skeleton.

Comparison of morphological features of *Jianianhualong* with those of other troodontids indicates that *Jianianhualong* has forelimbs and a pelvis closely resembling those of basal troodontids, but a cranium and hindlimbs that are more similar to those of derived troodontids. For example, *Jianianhualong* has forelimb and pelvic features seen in basal troodontids such as *Anchiornis* and *Sinovenator*[29,30] that include a humerus with a long inner tuberosity and short deltopectoral crest (also present in basal dromaeosaurids), a long manus (also present in some basal deinonychosaurs such as *Yixianosaurus*), a proportionally long metacarpal II, a proportionally long manual phalanx III-2 (longer than metacarpal III), a relatively small ilium (also in other basal paravians) with a wide, deep and slightly posteriorly directed pubic peduncle (also present in other derived maniraptorans), a posteroventrally oriented pubis, a pubic apron in which the dorsal portion along the midline extends posteriorly to form a longitudinal ridge, and a short ischium with posterodorsal and posteroventral processes and a distally positioned obturator process (also present in other basal paravians). Many cranial and hindlimb features of *Jianianhualong* are similar to those of derived troodontids, including a maxillary ascending process that extends posteriorly considerably beyond the anterior border of the antorbital fenestra, a dorsoventrally deep jugal ramus of the maxilla, a deep and narrow groove on the lateral surface of the jugal ramus of the maxilla, a sub-triangular antorbital fossa with a dorsally displaced anterior margin, the lack of a laterally visible promaxillary fenestra, an anteroventrally positioned maxillary fenestra that is significantly elongate anteroposteriorly, a relatively short jugal process of the postorbital, a proportionally long pedal digit I (phalanges I-1 and I-2 combined/metatarsal III length ratio 0.31), a metatarsal I positioned three-fifths of the way down the metatarsus, a proportionally long pedal phalanx I-1, a metatarsal II that is much more slender and considerably shorter than the other two main metatarsals, and a highly specialized pedal digit II as indicated by phalanx II-1 being $\sim 150\%$ the length of phalanx II-2, pedal phalanx II-2 having a proximoventral heel extending considerably both proximally and ventrally, and a pedal phalanx II-3 that is much larger than pedal phalanges II-1 and II-2.

The only other Jehol troodontid to display a mosaic of plesiomorphic and apomorphic osteological features is *Sinusonasus magnodens*, another troodontid with transitional anatomical features (Fig. 6). *Sinusonasus* has a cranium that closely resembles those of basal rather than derived troodontids and a pelvis and hindlimbs that are more similar to those of derived troodontids than to basal ones. For example, pelvic and hindlimb similarities shared by *Sinusonasus* and derived troodontids include an anteroventrally oriented pubis, an

elongate ischium with an obturator process at mid-length of the bone, and an arctometatarsalian metatarsus. Its cranial features that are shared with basal troodontids include a proportionately large and posteriorly extended external naris, a laterally exposed promaxillary fenestra, and a relatively short groove housing vascular foramina on the lateral surface of the dentary.

Natural selection has been suggested to act on modules rather than individual characters during major evolutionary transformations[72]. This hypothesis is supported by recent discoveries such as the pterosaur *Darwinopterus*, which has anatomical features that are transitional between the long-tailed basal pterosaurs and derived pterodactyloid pterosaurs[73]. Troodontidae is a small theropod family displaying a relatively low morphological disparity, but the distinct spatial organization of the mosaic of plesiomorphic and apomorphic osteological features in *Jianianhualong* and *Sinusonasus* raises the possibility of modular evolution in troodontids. The mosaic of plesiomorphic and apomorphic osteological features identified in *Jianianhualong* and *Sinusonasus* appears to show some correspondence to the expression domains of Hox genes, which more closely match the phenotypic modules identified in pterosaurs[73]. It seems possible that a functional driver may be involved in the mosaic of plesiomorphic and apomorphic osteological features identified in *Jianianhualong* and *Sinusonasus*, given their restriction to the skull, forelimbs and hindlimbs. In investigating the possibility of this driver, as well as the genetic regulation of these mosaics of plesiomorphic and apomorphic osteological features, there is a valuable opportunity to compare and contrast the mode of evolution along the different paravian lineages, particularly in understanding the early evolution of birds where aspects of previously proposed modular evolution have been supported[74]. Admittedly, the mosaic of plesiomorphic and apomorphic osteological characters observed here does not have as clear a distribution into modules as in species with transitional anatomical features that belong to major evolutionary transitions. For example, the cranium of *Jianianhualong* also displays some features seen in basal troodontids but absent in derived troodontids, including the large antorbital fenestra (as in *Sinovenator*, more than half the size of the antorbital fossa, and unlike the proportionally smaller one in *Byronosaurus*), a long maxillary tooth row terminating close to the posterior end of the jugal ramus of the maxilla, and most teeth have only relatively fine posterior serrations. The forelimb of *Jianianhualong* is similar to derived troodontids in being short and slender, and pedal digit II of *Sinusonasus magnodens* is not as specialized as in derived troodontids. Therefore, it remains a possibility that the mosaic of plesiomorphic and apomorphic osteological characters observed here, despite their distinct spatial organization, might have been produced by independent evolution of characters rather than the concerted evolution of characters within modules. Therefore, further study of mosaics of plesiomorphic and apomorphic osteological characters across paravians will be crucial towards deepening understanding of evolution along the different paravian lineages.

## Methods

**Material and its provenance and access.** DLXH 1218 is the holotype of *Jianianhualong tengi* and comprises of a nearly complete skeleton with associated feathers (Fig. 1). It was collected in Lower Cretaceous Yixian Formation rocks[12] of Baicai Gou (白菜沟), Yixian County, western Liaoning, China and is housed at the Dalian Xinghai Museum, Liaoning in accordance with local regulations. The specimen is available for public viewing and scientific study.

**Fossil excavation and preparation.** *Jianianhualong tengi* DLXH 1218 was excavated and prepared using standard methods. Excavation involved geological hammers, chisels and protective goggles and gloves. Preparation

was carried out using pneumatic tools in a preparation chamber fitted with a stereomicroscope.

**Comparative anatomical studies.** Standard comparative anatomy methods were used to study the new specimen. This involved comparing *Jianianhualong* with other fossil specimens either through first-hand studies (using a stereomicroscope to examine finer details) or literature-based studies.

**Laser-stimulated fluorescence imaging.** LSF imaging helped to clarify preserved feathering and osteology and confirmed the specimen's authenticity[52]. The imaging involved using a custom-built laser module (405 nm 500 mw laser) to illuminate the specimen with a raster-scanned laser line (produced by a Laserline Optics Canada lens) and then capturing the fluorescence produced in a long exposure shot, using a Nikon D610 DSLR camera fitted with a laser-blocking filter. For details of the LSF protocol see refs 52,75.

**Phylogenetic analysis.** The phylogenetic analysis was performed in the TNT software package[76] using a recently published dataset for coelurosaurian phylogeny[77] with *Jianianhualong* and *Sinusonasus* added in. This was run using a traditional search strategy with default settings other than setting the maximum trees in memory at 300,000 with 1,000 replications. Figure 6 shows the strict consensus of 3,066 most parsimonious trees (tree length = 1,426, CI = 0.279) produced by the analysis.

**Ancestral state reconstructions.** Ancestral state reconstructions for paravian arm and tail feather symmetry were performed in the evolutionary analysis software *Mesquite*[78] using the program's 'parsimony ancestral state reconstruction method' and the tree topology presented in Fig. 6.

**Measurement information.** Deltopectoral crest length is measured from the proximal extremity of the humerus to the summit point of the deltopectoral crest.

**Nomenclatural acts.** This published work and the nomenclatural acts it contains have been registered in ZooBank, the proposed online registration system for the International Code of Zoological Nomenclature. The ZooBank life science identifiers can be resolved and the associated information viewed by appending the life science identifiers to the prefix http://zoobank.org/. The life science identifiers for this publication are urn:lsid:zoobank.org:pub:1549DC59-3192-4834-9271-A8DB610969E1 and urn:lsid:zoobank.org:act:75D21EFF-2A12-45DE-85F9-7EABAF8CC6D3.

**Data availability.** The data reported in this paper are detailed in the main text and in the Supplementary Information.

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

## Acknowledgements

We thank Fangfang Teng for allowing us to study the specimen, Gao Xia for logistics during X.X.'s visit to the collection and Maureen Walsh for preparing the specimen. We also thank Thomas G. Kaye for his help in processing the LSF images. This study was supported by the National Science Foundation of China (41688103, 41120124002 and 91514302), the Dr Stephen S.F. Hui Trust Fund (201403173007), the Research Grant Council of Hong Kong's General Research Fund (17103315) and the Faculty of Science of the University of Hong Kong.

## Author contributions

X.X., P.C. and M.P. designed the project, X.X., P.C., M.P., L.X., Q.M., J.L. and C.Y. performed the research, and X.X., P.C. and M.P. wrote the manuscript.

## Additional information

**Competing interests:** The authors declare no competing financial interests.

