## [Peer Review File · Nature Communications]

Reviewers' comments:

Reviewer #1 (Remarks to the Author):

This is a very well-done paper that describes a new troodontid with asymmetrical feathers. Although such feathers have been reported before, this confirms their presence in troodontids. Troodontids are not birds but are close to them; they did not fly, so the question is why their feathers were symmetrical. The ability to glide is suggested here but not supported; asymmetrical feathers occur in other non-avian dinosaurs that did not fly, and appear today in some birds even though they are not used in flying. So their presence does not automatically relate to any kind of flight. It is known from developmental studies by Prum and his colleagues that asymmetry develops from symmetry ontogenetically. How often this has happened in maniraptorans is fundamental to know before one can assess the meaning or importance of this pattern, and for this a more detailed phylogenetic analysis of the distribution of asymmetrical feathers would be needed.

The description is very competent, although also very long, and most of it is of interest only to specialists, so it could be put in supplemental information.

Throughout the manuscript: it is unnecessary to say "large in size" or "small in size." "Large" and "small" are sufficient.

Also: evolutionary biologists who work on functional transitions in major taxa do not generally like to use the term "transitional form" because it implies that such a "transitional form" is a direct ancestor or link between two other taxa, and we don't like to claim or imply that we're finding or looking for "direct ancestors" in the fossil record. So it is better to talk about taxa that show transitional features.

The term "modularity" is fashionable but over-used. It is good to be very specific about what is considered "modular" and why. Frequently, authors speak about functional modules, as Gatesy and Dial did in discussing the forelimb, hindlimb, and tail units that evolved independently in basal birds, considering shifts in functions and dominance. One can also discuss developmental modules, as has been done for the basal tetrapod limb, especially considering the expression of specific genes and pathways in the formation and configuration of digits. Here, there appears to be some variation in expression of feather asymmetry among several lines of basal maniraptorans, but this is not convincingly tied to function or development, and there is insufficient phylogenetic context to know how many times certain features evolved in the clade – consequently, it is not clear what is "modular" as opposed to simply "variable." This confusion is reflected in the authors' statement that "their differing modularities demonstrate that modular evolution is variable even during the evolutionary transitions of small theropod clades."

This is already a very good paper, and I would only suggest that the authors may wish to give a bit more thought to the conceptual framework of what they describe as "modularity."

Reviewer #2 (Remarks to the Author):

The manuscript entitled "Modular evolution and feather asymmetry in a transitional feathered troodontid (Dinosauria: Theropoda)" by Dr. Xu and colleagues describes for the first time a definitive troodontid with the feather plumage preserved. This is a very important discovery as the phylogenetic position of many other potential basal troodontids has been questioned several times. Not surprising, Jianianhualong resembles other Pennaraptora in having pennaceous feathers, but their morphology and distribution (tail and hind limb feathers in particular) helps to understand the variability of the plumage in these very bird-like theropods in more detail. Therefore, this aspect should be a little bit more extended in the paper. As I argue below, the authors should develop for instance a hypothesis for an alternative biological role of the asymmetric rectrices, if the animal could not fly. Besides, it would be stated how the new find fits in current evolutionary concepts of hind limb feathering in Pennaraptora. Otherwise the paper is well written and straight forward. The description is very concise, highlighting the most important anatomical features. The comparison with other taxa could be sometimes a little bit more detailed (although I am aware of the word limit in the journal). However, I would appreciate, if at least *Mei long* and *Sinusonasus* could be stronger included in the comparison. In sum, I recommend minor revisions of the manuscript and fully support its publication in *Nature Communication* with no doubts. Some more minor points, however, that should be addressed are listed below:

- P2L49 and P18L361: The authors state that the troodontid affinities of *Anchiornis*, *Eosinopteryx*, *Xiaotingia* and *Jinfengopteryx* were questioned by several authors (ref. 3 and 17). In addition, this was also challenged by Foth et al. 2014 (ref. 18) and to some degree by Xu et al. 2011 (ref. 6).
- P2L30 and P19L365: I am not really happy with the use of the term "four-winged". The authors are correct that Paraves/Eumaniraptora had elongated pennaceous feathers along the tibia, resembling the condition of recent birds of prey for instance. However, one would not call an eagle to be "four-winged" just because of its tibial feather breeches. In consequence, I would restrict the term 'four-winged' only if the metatarsal region is included in extensive feathering, as it is the case in *Anchiornis* or *Microraptor*.
- P4L79: I think that the "high, sub-triangular cranium" is a little vague as a diagnostic character, because it also occurs in other small-bodied theropods (including *Archaeopteryx*). No proportions are given to justify this character. Besides, the skull is heavily crushed so that the original shape may be hard to reconstruct. As the authors present a pretty long list of other diagnostic characters, they may delete the character mentioned.
- P4L83f: May reword: "a distinct fossa on the dorsal surface of the surangular close to the posterior end".
- P7L118f: Please, change "ventral ramus" to "jugal ramus". This is more figurative.

- P7L123ff: A promaxillary fenestra is also present in Sinusonasus, Anchiornis and Jinfengopteryx.
- P7L124ff: May reword to: "The maxillary fenestra is enlarged and significantly elongate anteroposteriorly as in other troodontids (particularly derived ones^{28,30}), ..." Besides, the diversity of elongated maxillary fenestrae in troodontids is well illustrated in Senter et al. (2010, PLoS ONE 5: e14329).
- P8L140: Do the foramina along the lateral edge result from pneumaticity (as some allosauroids) or vascularity?
- P8L146: In the introduction of the cranial description the authors say that the parietals are preserved "upside down". If the parietals show the ventral aspect (as the frontals), how can anything said about the morphology of the sagittal crest?
- P8L153f: May add the description of the morphology of the head and the pterygoid process (see suppl. information).
- P8L153f: If the morphology of the pterygoid is not described here in any detail (see suppl. information), the sentence can be deleted.
- P9L169: On which specimen of Sinovenator the authors refer their comparison? As described (ref. 9) only the dentary is preserved in the holotype of Sinovenator.
- P9L172: I am not a native speaker, but should there be a "that" between "show" and "this"?
- P9L192ff: The authors may add a reference to the suppl. information to point to the more detailed description of the vertebral column presented their.
- P11L219ff: Could the authors state something on the morphology of the furcula, glenoid fossa and the acromion in the main text?
- P13L238f: A bowed ulnae seem to be also present in Mei. This might be a useful comparison.
- P15L283ff: I would appreciate, if the authors could mark the dorsal posterior process of the ischium in Fig. 4. I cannot recognize it from the photos due to the overlap with other bones. Furthermore, it might be worth to mention that a prominent ventral posterior process is also present in Archaeopteryx. Last but not least, I would like to know what the authors mean with "medial lamina" in the obturator process and how is this visible, if medially located?
- P17L333: Long pennaceous rectrices along the entire tail seem to be also present in Anchiornis (see Lindgren 2015, Sci. Rep. 5: 13520). Please include for comparison.

- P18L337: Asymmetric pennaceous rectrices are also present in Archaeopteryx (see ref. 18). Please include for comparison.
- P18L346: When the authors list all the troodontid affinities they may also include the lateral ridge along the jugal.
- P19L382f: Regarding poor feather preservation and the impact on the interpretation of the morphology they authors may also cite Foth (2012, Pal Z 86:91-102).
- P19L383ff: How can the authors be sure that only the tail, but not the wing feathers were asymmetric in shape? The preservation does not really allow this conclusion, right? One could argue, because of the short forelimbs it is unlikely that Jianianhualong used the wings for aerial locomotion, and therefore the primaries were probably not asymmetric. But, if so the authors should give a possible explanation, why the tail feathers (and not the wing feathers) show this particular morphology. Could it have an impact on cursorial locomotion, indicating the importance of a (shorter) tail to control the balance? Maybe a little bit speculation should be included here. Otherwise, this interesting plumage feature eclipses a little bit. Furthermore, the authors may add a statement on the distribution of the hind limb feathers, as Jianianhualong seem to support the mosaic evolution of the plumage in this region (see Foth et al. 2014; ref. 18).
- P22L433ff: This paragraph is slightly confusing in his order regarding derived and non-derived features. I suggest rewriting this part a little bit.
- P22L444: Change to: Natural selection
- Is it possible to illustrate the modularity of derived and non-derived characters of the Jianianhualong (and maybe Sinusonasus)? Besides pterosaurs, are there other examples in vertebrate palaeontology, which confirm modular evolution?
- Fig. 1 and Fig. 4. Something seems to be wrong regarding the labeling of the scapulae and humeri. What is labeled as left scapula in Fig. 4a looks like the left humerus to me.
- I would appreciate if the authors could provide measurements in the supplementary information

Christian Foth

Reviewer #3 (Remarks to the Author):

Dear Editor,

After a detailed overview of the manuscript, I think that it constitutes a novel and very

important contribution to vertebrate paleontology. The manuscript is of broad interest for paleontologists and biologists interested in the early origin and evolution of birds.

It is very well-written and constitutes an important addition for the knowledge of feathers in non-avian theropods.

The manuscript should be accepted with minor changes. I am also attaching a PDF of the reviewed version of the manuscript with additional comments.

Everything else in which I can help you, let me know,

All the best,

Federico Agnolin
LACEV, MACN

Xu, Currie, Pittman *et al.* response to Reviewer #1

We would like to thank Reviewer #1 for the positive and helpful review. We are glad they thought our paper was ‘very well-done’. We have made edits to our manuscript based on all of their suggestions.

This is a very well-done paper that describes a new troodontid with asymmetrical feathers. Although such feathers have been reported before, this confirms their presence in troodontids. Troodontids are not birds but are close to them; they did not fly, so the question is why their feathers were symmetrical. The ability to glide is suggested here but not supported; asymmetrical feathers occur in other non-avian dinosaurs that did not fly, and appear today in some birds even though they are not used in flying. So their presence does not automatically relate to any kind of flight. It is known from developmental studies by Prum and his colleagues that asymmetry develops from symmetry ontogenetically. How often this has happened in maniraptorans is fundamental to know before one can assess the meaning or importance of this pattern, and for this a more detailed phylogenetic analysis of the distribution of asymmetrical feathers would be needed.

The authors have been careful not to imply gliding abilities in *Jianianhualong* by avoiding the term in the manuscript. Instead we use ‘aerodynamic capabilities’ and ‘aerodynamic prowess’ which are terms that are not limited to the movement of feathered dinosaurs through the air, but can also apply to terrestrial locomotion. We find great value in producing ancestral state reconstructions of wing and tail feather symmetry across the paravian tree. We have done this using the topology recovered in figure 6 and have included additional discussion points based on these new data to support the great significance of asymmetrical tail feathers in *Jianianhualong*. We would like to especially thank Reviewer #1 for this important suggestion.

The description is very competent, although also very long, and most of it is of interest only to specialists, so it could be put in supplemental information.

We have reviewed the description taking on board Reviewer #1’s suggestion to shorten the main text description and Reviewer #2’s (Dr. Foth’s) suggestion to lengthen it. The description in the main text now allows the reader to consider the discussion section without the need to frequently refer to Supplemental Note 2. Supplemental Note 2 contain details that are less important towards the discussion section and the average reader.

Throughout the manuscript: it is unnecessary to say “large in size” or “small in size.” “Large” and “small” are sufficient.

As suggested, ‘in size’ has been removed from the relevant places in the manuscript.

Also: evolutionary biologists who work on functional transitions in major taxa do not generally like to use the term “transitional form” because it implies that such a “transitional form” is a direct ancestor or link between two other taxa, and we don’t like to claim or imply that we’re finding or looking for “direct ancestors” in the fossil record. So it is better to talk about taxa that show transitional features.

Thank you for this comment. As suggested, we have adjusted our usage of ‘transitional’ to more clearly correspond to anatomical features rather than whole animals.

The term “modularity” is fashionable but over-used. It is good to be very specific about what is considered “modular” and why. Frequently, authors speak about functional modules, as Gatesy and Dial did in discussing the forelimb, hindlimb, and tail units that evolved

independently in basal birds, considering shifts in functions and dominance. One can also discuss developmental modules, as has been done for the basal tetrapod limb, especially considering the expression of specific genes and pathways in the formation and configuration of digits. Here, there appears to be some variation in expression of feather asymmetry among several lines of basal maniraptorans, but this is not convincingly tied to function or development, and there is insufficient phylogenetic context to know how many times certain features evolved in the clade – consequently, it is not clear what is “modular” as opposed to simply “variable.” This confusion is reflected in the authors’ statement that “their differing modularities demonstrate that modular evolution is variable even during the evolutionary transitions of small theropod clades.”

The implications of the asymmetrical tail feathers of *Jianianhualong* are discussed separately to the skeletal modularity observed in *Jianianhualong* and *Sinuso nasus*. Reviewer #1 appears to have accidentally mixed these two discussion points together in providing the comments in the paragraph above. As the authors discussed these two points separately, we feel that the modular distribution of osteological features that we observed in the skeletons of *Jianianhualong* and *Sinuso nasus* follow the framework adopted by existing studies discussing modular evolution in other fossil vertebrates and is reasonable. We acknowledge the need to understand the genetic basis of modular evolution and its possible ties to function and have noted this in the manuscript.

This is already a very good paper, and I would only suggest that the authors may wish to give a bit more thought to the conceptual framework of what they describe as “modularity.”

Please see our response above regarding modularity.

Xu, Currie, Pittman *et al.* response to Reviewer #2

We would like to thank Dr. Foth for his helpful and positive review. We are happy to hear that our paper presents ‘a very important discovery’. We have made edits based on all of his suggestions.

The manuscript entitled “Modular evolution and feather asymmetry in a transitional feathered troodontid (Dinosauria: Theropoda)” by Dr. Xu and colleagues describes for the first time a definitive troodontid with the feather plumage preserved. This is a very important discovery as the phylogenetic position of many other potential basal troodontids has been questioned several times. Not surprising, *Jianianhualong* resembles other Pennaraptora in having pennaceous feathers, but their morphology and distribution (tail and hind limb feathers in particular) helps to understand the variability of the plumage in these very bird-like theropods in more detail. Therefore, this aspect should be a little bit more extended in the paper. As I argue below, the authors should develop for instance a hypothesis for an alternative biological role of the asymmetric rectrices, if the animal could not fly. Besides, it would be stated how the new find fits in current evolutionary concepts of hind limb feathering in Pennaraptora. Otherwise the paper is well written and straight forward. The description is very concise, highlighting the most important anatomical features. The comparison with other taxa could be sometimes a little bit more detailed (although I am aware of the word limit in the journal). However, I would appreciate, if at least *Mei long* and *Sinuso nasus* could be stronger included in the comparison. In sum, I recommend minor revisions of the manuscript and fully support its publication in Nature Communication with no doubts. Some more minor points, however, that should be addressed are listed below:

As suggested, the authors have noted the significance of elongated tibial feathers in *Jianianhualong* towards the evolution of paravian hind limb feathering. The authors have also included additional text on the biological role of the asymmetric rectrices in this animal, as suggested. Additional anatomical comparisons featuring *Mei long* and *Sinuserosaurus* have been added, as requested. We have reviewed the description taking on board Reviewer #1's suggestion to shorten the main text description and Reviewer #2's (Dr. Foth's) suggestion to lengthen it. The description in the main text now allows the reader to consider the discussion section without the need to frequently refer to Supplemental Note 2. Supplemental Note 2 contain details that are less important towards the discussion section and the average reader.

•P2L49 and P18L361: The authors state that the troodontid affinities of *Anchiornis*, *Eosinopteryx*, *Xiaotingia* and *Jinfengopteryx* were questioned by several authors (ref. 3 and 17). In addition, this was also challenged by Foth et al. 2014 (ref. 18) and to some degree by Xu et al. 2011 (ref. 6).

As suggested, we have added two additional citations to support this point.

•P2L30 and P19L365: I am not really happy with the use of the term “four-winged”. The authors are correct that Paraves/Eumaniraptora had elongated pennaceous feathers along the tibia, resembling the condition of recent birds of prey for instance. However, one would not call an eagle to be “four-winged” just because of its tibial feather breeches. In consequence, I would restrict the term ‘four-winged’ only if the metatarsal region is included in extensive feathering, as it is the case in *Anchiornis* or *Microraptor*.

We understand Dr. Foth's concern, but we would like to retain the use of this term for the following reasons. Firstly, because of convenience for the description and secondly, because the condition in basal paravians is different from the condition in some recent raptors in that the former always display a coherent surface by the leg feathers. Finally, usage of this term in the text has no effect on the conclusions drawn in the present paper.

•P4L79: I think that the “high, sub-triangular cranium” is a little vague as a diagnostic character, because it also occurs in other small-bodied theropods (including *Archaeopteryx*). No proportions are given to justify this character. Besides, the skull is heavily crushed so that the original shape may be hard to reconstruct. As the authors present a pretty long list of other diagnostic characters, they may delete the character mentioned.

As suggested, the diagnostic character ‘high, sub-triangular cranium’ has been removed.

•P4L83f: May reword: “a distinct fossa on the dorsal surface of the surangular close to the posterior end”.

This sentence has been reworded, as suggested.

•P7L118f: Please, change “ventral ramus” to “jugal ramus”. This is more figurative. ‘Ventral ramus’ of the maxilla has been changed to ‘jugal ramus’, as suggested. The first use of ‘jugal ramus’ is preceded by ‘(ventral)’ for the benefit of workers that are more familiar with the term ‘ventral ramus’.

•P7L123ff: A promaxillary fenestra is also present in *Sinuserosaurus*, *Anchiornis* and *Jinfengopteryx*.

These examples have been added, as suggested.

•P7L124ff: May reword to: “The maxillary fenestra is enlarged and significantly elongate anteroposteriorly as in other troodontids (particularly derived ones 28,30), ...” Besides, the

diversity of elongated maxillary fenestrae in troodontids is well illustrated in Senter et al. (2010, PLoS ONE 5: e14329).

As suggested, this sentence has been reworded and an additional citation has been added.

•P8L140: Do the foramina along the lateral edge result from pneumaticity (as some allosauroids) or vascularity?

We are not sure but this would be an interesting question to answer using troodontids CT data. However, this work is beyond the scope of this study.

•P8L146: In the introduction of the cranial description the authors say that the parietals are preserved “upside down”. If the parietals show the ventral aspect (as the frontals), how can anything said about the morphology of the sagittal crest?

Thanks for spotting this. The parietals are actually the right way up so this has been corrected in the introduction of the cranial description.

•P8L153f: May add the description of the morphology of the head and the pterygoid process (see suppl. information).

This description has been added, as suggested.

•P8L153f: If the morphology of the pterygoid is not described here in any detail (see suppl. information), the sentence can be deleted.

This sentence has been deleted, as suggested.

•P9L169: On which specimen of *Sinovenator* the authors refer their comparison? As described (ref. 9) only the dentary is preserved in the holotype of *Sinovenator*.

The specimen we refer to is IVPP V20378. This specimen number has now been added to the text.

•P9L172: I am not a native speaker, but should there be a “that” between “show” and “this”?

This edit has been made, as suggested.

•P9L192ff: The authors may add a reference to the suppl. information to point to the more detailed description of the vertebral column presented their.

This reference has been added, as suggested.

•P11L219ff: Could the authors state something on the morphology of the furcula, glenoid fossa and the acromion in the main text?

This text has been added, as suggested.

•P13L238f: A bowed ulnae seem to be also present in *Mei*. This might be a useful comparison.

The comparison has been added, as suggested.

•P15L283ff: I would appreciate, if the authors could mark the dorsal posterior process of the ischium in Fig. 4. I cannot recognize it from the photos due to the overlap with other bones. Furthermore, it might be worth to mention that a prominent ventral posterior process is also present in *Archaeopteryx*. Last but not least, I would like to know what the authors mean with “medial lamina” in the obturator process and how is this visible, if medially located?

The posterior processes of the ischium have been labelled, as suggested. In addition, the prominent ventral posterior process of *Archaeopteryx* has been mentioned. The term medial lamina refers to a medially-aligned lamina. It is possible to remove the term 'medial' and still preserve the meaning of the sentence so we have done this to improve the clarity of the text.

•P17L333: Long pennaceous rectrices along the entire tail seem to be also present in *Anchiornis* (see Lindgren 2015, Sci. Rep. 5: 13520). Please include for comparison.
This example has been added to the comparison, as suggested.

•P18L337: Asymmetric pennaceous rectrices are also present in *Archaeopteryx* (see ref. 18). Please include for comparison.
This example has been added to the text, as suggested.

•P18L346: When the authors list all the troodontid affinities they may also include the lateral ridge along the jugal.
The list of troodontid affinities has been reviewed by the authors accordingly.

•P19L382f: Regarding poor feather preservation and the impact on the interpretation of the morphology they authors may also cite Foth (2012, Pal Z 86:91-102).
This citation has been added, as suggested.

•P19L383ff: How can the authors be sure that only the tail, but not the wing feathers were asymmetric in shape? The preservation does not really allow this conclusion, right? One could argue, because of the short forelimbs it is unlikely that *Jianianhualong* used the wings for aerial locomotion, and therefore the primaries were probably not asymmetric. But, if so the authors should give a possible explanation, why the tail feathers (and not the wing feathers) show this particular morphology. Could it have an impact on cursorial locomotion, indicating the importance of a (shorter) tail to control the balance? Maybe a little bit speculation should be included here. Otherwise, this interesting plumage feature eclipses a little bit. Furthermore, the authors may add a statement on the distribution of the hind limb feathers, as *Jianianhualong* seem to support the mosaic evolution of the plumage in this region (see Foth et al. 2014; ref. 18).
Dr. Foth's points have been taken on board and the text has been edited accordingly.

•P22L433ff: This paragraph is slightly confusing in his order regarding derived and non-derived features. I suggest rewriting this part a little bit.
This paragraph has been rewritten, as suggested.

•P22L444: Change to: Natural selection
This has been changed, as suggested.

•Is it possible to illustrate the modularity of derived and non-derived characters of the *Jianianhualong* (and maybe *Sinusoasus*)? Besides pterosaurs, are there other examples in vertebrate palaeontology, which confirm modular evolution?
The authors feel that the description is sufficient without an illustration since the spatial relationship of the anatomical features concerned are obvious. Other examples exist in vertebrate palaeontology including in rodents, but we selected pterosaurs as our main example and have added a key citation for modular evolution in birds because of their greater phylogenetic relevance to troodontids.

•Fig. 1 and Fig. 4. Something seems to be wrong regarding the labelling of the scapulae and humeri. What is labelled as left scapula in Fig. 4a looks like the left humerus to me.
This was an error and has been corrected. Good spot!

•I would appreciate if the authors could provide measurements in the supplementary information
These have been provided.

Xu, Currie, Pittman *et al.* response to Reviewer #3

We would like to thank Dr. Agolin for his extremely positive review - we were thrilled to hear that the manuscript 'constitutes a novel and very important contribution to vertebrate paleontology'. We have made edits to the manuscript based on all of his suggestions.

Dear Editor,

After a detailed overview of the manuscript, I think that it constitutes a novel and very important contribution to vertebrate paleontology. The manuscript is of broad interest for paleontologists and biologists interested in the early origin and evolution of birds. It is very well-written and constitutes an important addition for the knowledge of feathers in non-avian theropods.

The manuscript should be accepted with minor changes. I am also attaching a PDF of the reviewed version of the manuscript with additional comments.
Everything else in which I can help you, let me know,

All the best,

Federico Agnolin
LACEV, MACN

REVIEWERS' COMMENTS:

Reviewer #2 (Remarks to the Author):

The revised version of the manuscript 'Modular evolution in an asymmetrically feathered troodontid (Dinosauria: Theropoda) with transitional features' by Xu and colleagues has been significantly improved. All points (also from the other reviewers have been adequately addressed). The description part is straight, but very informative (and should not be shortened). Reading it again I found only some minor issues that should be corrected:

- Pretty minor: Could the authors provide a long version for IVPP (in context of Sinovenator).
- The authors should decide if they want to use species (e.g. Anchiornis huxelyi) or just genera names (Anchiornis). This inconsistency can be found for various taxa and in both main text and supplementary information.
- P9L379f: The statement is correct that outside Avialae the only other clade for which asymmetric pennaceous feathers are documented are Microraptorinae. But is this also the case for other basal Dromaeosauridae, e.g. Sinornithosaurus and Zhenyuanlong?
- P10L414f: This is a justified speculation. Accordingly, the authors should indicate this by rephrasing the introducing sentence 'It is worth to mention ...'. I think that 'mention' is not the adequate word in this context. Maybe 'hypothesize', 'speculate', 'presume' or something in this direction.
- P10L422ff: The last paragraph of this section does not really fit thematically. In my opinion, it would be good introducing paragraph for the following section (Modular evolution in troodontids).

Christian Foth

Reviewer #3 (Remarks to the Author):

Dear Editor,

The authors have been constested in detail and properly. They have followed most suggestion made by the referees.

In this way, I think that the MS may be accepted as it is, without additional corrections,

All the best,

Federico Agnolin

Reviewer #2 (Remarks to the Author):

The revised version of the manuscript ‘Modular evolution in an asymmetrically feathered troodontid (Dinosauria: Theropoda) with transitional features’ by Xu and colleagues has been significantly improved. All points (also from the other reviewers have been adequately addressed). The description part is straight, but very informative (and should not be shortened). Reading it again I found only some minor issues that should be corrected:

- Pretty minor: Could the authors provide a long version for IVPP (in context of *Sinovenator*).

-Edited as requested.

- The authors should decide if they want to use species (e.g. *Anchiornis huxelyi*) or just genera names (*Anchiornis*). This inconstancy can be found for various taxa and in both main text and supplementary information.

-We has reviewed this aspect and have clarified taxon names where appropriate.

- P9L379f: The statement is correct that outside Avialae the only other clade for which asymmetric pennaceous feathers are documented are Microraptorinae. But is this also the case for other basal Dromaeosauridae, e.g. *Sinornithosaurus* and *Zhenyuanlong*?

-Unfortunately, other basal dromaeosaurids don’t preserve their feathering sufficiently to determine their asymmetry (including *Sinornithosaurus* and *Zhenyuanlong*?).

- P10L414f: This is a justified speculation. Accordingly, the authors should indicate this by rephrasing the introducing sentence ‘It is worth to mention ...’. I think that ‘mention’ is not the adequate word in this context. Maybe ‘hypothesize’, ‘speculate’, ‘presume’ or something in this direction.

-Edited as requested.

- P10L422ff: The last paragraph of this section does not really fit thematically. In my opinion, it would be good introducing paragraph for the following section (Modular evolution in troodontids).

-Edited as requested.

Reviewer #3 (Remarks to the Author):

Dear Editor,

The authors have been constested in detail and properly. They have followed most suggestion made by the referees.

In this way, I think that the MS may be accepted as it is,without additional corrections,

All the best,

Federico Agnolin

We would like to thank the reviewers for their valuable feedback.